# Bacillamide D produced by *Bacillus cereus* from the mouse intestinal bacterial collection (miBC) is a potent cytotoxin in vitro
Maximilian Hohmann[1], Valentina Brunner[2,3,4], Widya Johannes[5], Dominik Schum [6], Laura M. Carroll[7], Tianzhe Liu[1], Daisuke Sasaki[5,8], Johanna Bosch [9], Thomas Clavel [9], Stephan A. Sieber [6], Georg Zeller [7], Markus Tschurtschenthaler [2,3,4,11] ✉, Klaus-Peter Janßen [5,11] ✉ & Tobias A. M. Gulder [1,10,11] ✉

The gut microbiota influences human health and the development of chronic diseases. However, our understanding of potentially protective or harmful microbe-host interactions at the molecular level is still in its infancy. To gain further insights into the hidden gut metabolome and its impact, we identified a cryptic non-ribosomal peptide BGC in the genome of *Bacillus cereus* DSM 28590 from the mouse intestine (www.dsmz.de/miBC), which was predicted to encode a thiazol(in)e substructure. Cloning and heterologous expression of this BGC revealed that it produces bacillamide D. In-depth functional evaluation showed potent cytotoxicity and inhibition of cell migration using the human cell lines HCT116 and HEK293, which was validated using primary mouse organoids. This work establishes the bacillamides as selective cytotoxins from a bacterial gut isolate that affect mammalian cells. Our targeted structure-function-predictive approach is demonstrated to be a streamlined method to discover deleterious gut microbial metabolites with potential effects on human health.

The mammalian gut environment hosts complex communities of microorganisms that are essential for maintaining host health and can also influence disease development and/or progression[1–4]. In humans, gut microbiota alterations are linked, for example, to colorectal cancer (CRC) and inflammatory bowel disease (IBD). While there has been considerable progress in understanding these diseases, their etiology and specific contributions of gut microbes remain incompletely understood[5,6].

Bacteria are well-known producers of functional small molecules not directly involved in primary metabolism. In contrast to virulence factors such as Shiga toxin from *Shigella* and pathogenic *E. coli* species, which have been extensively studied in the context of human pathologies and, in particular, tumorigenesis[7–9], these specialized compounds, termed natural products (NPs), are still barely understood. NPs are genetically encoded, with the required biosynthetic genes typically organized in close genetic proximity within biosynthetic gene clusters (BGCs)[10]. NPs fulfill various biological functions for their producers, ranging from intra- and inter-species communication to chemical warfare in the competition for scarce nutrients and ecological niches. Bacterial NPs have a long and successful history as drugs and lead structures in human medicine[11], such as for antibiotics (e.g., vancomycin, tetracyclines, and erythromycins), immunosuppressants (e.g., cyclosporins) and anti-cancer chemotherapeutics (e.g., mitomycins, doxorubicin, and epothilones)[12]. However, de novo-produced NPs from different members of complex microbial ecosystems, such as the human gut microbiome, and their potential effects on host cells have remained poorly understood.

Given their potent biological effects, NPs are likely to play decisive roles in microbe-microbe interactions, which in turn shape microbiome composition and function. As NP-mediated negative microbe-microbe interactions contribute to colonization resistance or dysbiosis (e.g., by action of specific antimicrobials), they can directly affect host health[13,14]. Additionally, NPs can even have direct effects on disease development and progression, e.g., by inducing cytotoxicity or mutagenicity. A very limited—yet functionally highly relevant—number of examples of microbially synthesized NPs with such pathophysiological effects on the host are known: the non-ribosomal peptide (NRP) tilivalline (**1**), produced by the gut bacterium *Klebsiella oxytoca*[15], was shown to have enterotoxic activity and to induce

apoptosis in human cells in vitro. However, a clear correlation between the producing strain or the tilivalline-encoding BGC and the occurrence of IBD in patients could not be shown[16]. Colibactin (**2**) is an NRP-polyketide-hybrid found in genotoxic *E. coli* strains overrepresented in CRC patients[17]. The molecule has been shown to crosslink DNA and thereby has been hypothesized to cause CRC[18–20]. Indeed, it has been shown that prolonged exposure of human organoids to colibactin-producing *E. coli* strains leads to distinct mutational signatures also found in CRC[21]. However, while the colibactin BGC is found in about 60% of CRC cases, it is also identified in 20% of control samples[22].

Apart from these examples, the biosynthetic potential of mammalian gut bacteria, in particular, to produce deleterious NPs, has been understudied, with only a few other compounds identified so far, among them the DNA-damaging indolimines[23] and ribosomal peptides with antibiotic activity[24]. This highlights the need to accelerate NP discovery from gut bacteria, not only focusing on finding novel potential therapeutics for medical application but also uncover harmful metabolite-driven microbe-host interactions. Most genome mining efforts so far have been limited to the human microbiome[25,26]. However, mouse models have long been invaluable in the microbiome as well as biomedical research[27], and insights into the biosynthetic potential within the mouse gut microbiota could thus translate into the human gut. Substantial progress has been made in understanding, isolating, and culturing bacterial strains from the murine intestine, as demonstrated by the establishment of public resources such as the mouse intestinal bacterial collection (miBC)[3,28]. This enabled us to systematically analyze the genomes of isolates for BGCs encoding heterocyclic NPs, which led to the identification of two members of the bacillamide family as potent cytotoxic NPs.

## Results and discussion
### Identification of a BGC putatively encoding heterocyclic structural elements
The main obstacles in discovering gut microbial NPs potentially impacting host health are (i) the selection of the most promising BGCs to be studied out of the vast metabolic reservoir present and (ii) the production, isolation, and characterization of the encoded NPs. Various small molecules substantially impacting metabolic processes contain heterocyclic structural features that enable strong interactions with their targets, for example, DNA. In Nature, many such heterocyclic NPs are encoded by modular biosynthetic machinery, such as polyketide synthases (PKS) or NRP synthetases (NRPS)[29,30]. This is also the case for tilivalline (**1**) with a benzodiazepine-type substructure and for colibactin (**2**) containing thiazole structural elements, which are crucial for its DNA-binding and -crosslinking activity (Fig. 1)[19,31].

NRPSs biosynthetically fuse amino acid precursors in an assembly-line-like fashion[30]. Each biosynthetic module processes one building block. In each module, selective adenylation domains (A) activate the required precursor and attach it to individual peptidyl carrier proteins, also called thiolation domains (T). The selectivity of A domains can bioinformatically be predicted based on the Stachelhaus code[32–34]. The formation of heterocyclic features in NRPS pathways can usually be traced back to adenylation domains activating serine, cysteine, or threonine building blocks that contain nucleophilic side chains, enabling downstream cyclization chemistry. In addition, the respective BGCs contain domains promoting heterocyclization (cyclization domains: Cy) and, optionally, oxidative domains for aromatization, e.g., dehydrogenases (DH). As these NRPS features can readily be identified computationally, we set out to systematically screen the genomes of bacterial strains isolated from the mouse gut using antiSMASH[35,36] and Prism[37] to identify NRPSs putatively producing heterocyclic NPs. This led to the identification, among others (see Figure S1 for details), of a promising cryptic BGC, henceforth termed *bac*, from *B. cereus* DSM 28590. The *bac* BGC was selected because of its relatively small size, facilitating straightforward pathway interception and construction of a suitable expression vector for recombinant compound production. It consists of three genes on a single transcript in the following order: A 1 kb gene encoding a DH, a 7 kb gene encoding an NRPS- and a 1.5 kb gene

encoding a decarboxylase (DC). The translated NRPS protein comprises two biosynthetic modules, one consisting of an A-domain paired with a cyclization domain, predicted to activate cysteine, thus most likely forming a thiazol(in)e ring.

To unlock the metabolites produced by this BGC, we aimed at its interception by Direct Pathway Cloning (DiPaC)[38–40], with subsequent recombinant fermentative NP production. DiPaC is a streamlined in vitro cloning strategy relying on long-amplicon PCR using high-fidelity polymerases and homology-based assembly methods for rapidly cloning BGCs into any desirable expression vector. This approach is particularly efficient to clone small- to midsized BGCs of up to ~23 kb in a single step, such as the 9.5 kb *bac* BGC. In addition, DiPaC readily allows for reorganization and/or partial deletion of genetic sequences directly during the initial cloning procedure. Utilizing the ARNold tool[41], a putative rho-dependent transcriptional terminator sequence was identified in the intergenic region between the NRPS- and the DC-encoding genes. This 55 bp intergenic sequence was excluded from the expression construct by splitting the BGC into two amplicons, the first incorporating the DH- and NRPS-encoding genes, the second only the DC-encoding one. Reassembly of the two amplicons in the expression vector backbone pET28b-ptetO::*gfp* thus directly delivered the terminator-free expression plasmid (see Fig. 2a–c). This plasmid makes use of the ptetO promoter system, which is inducible with tetracycline and has been successfully used for activation of several NP pathways[38–40,42,43]. The *gfp*-gene downstream of the insert allows for a quick transcriptional control after expression, owing to the fluorescence of its product GFP[39].

### Production of the target NP in a heterologous host
The expression vector pET28b-ptetO::*bac*::*gfp* was produced by cultivating *E. coli* DH5α, purifying it, and transforming it into *E. coli* BAP1. This strain contains a chromosomal copy of *sfp*, the gene encoding the phospho-pantetheinyl transferase Sfp, which activates PKS/NRPS machinery by posttranslational activation of T domains[44]. Expression was induced with tetracycline, and the expression strain was cultivated for 5 days at 20 °C in terrific broth (TB medium) before the cells were harvested by centrifugation.

Extraction and HPLC analysis of the cell pellets revealed the presence of two new substances at 9.2 min and 12.1 min, which were not found in control cultures of *E. coli* BAP1 only containing the empty expression vector (Fig. 3a). These substances were found in even higher amounts in extracts of basified supernatants. Both substances were purified by preparative HPLC and characterized using 2D-NMR and High-resolution mass spectrometry (HRMS). These investigations revealed the compound that eluted at 9.2 min to be tryptamine (**3**). The slower eluting NP contained **3** as a biosynthetic building block and was identified as bacillamide D (**4**). To our knowledge, this marks the first complete heterologous production of a member of the bacillamide family, remarkably at a production titer of 77 mg/L, enabling a thorough examination of its biological effects. We furthermore confirmed that **4** is produced by the native *B. cereus* strain (Figure S8), albeit in miniscule amounts in the laboratory setting.

Bacillamide D (**4**) is indeed a thiazole-containing NP. Biosynthetically, the A domains of the NRPS system activate D-alanine (A₁) and D-cysteine (A₂) by their attachment to their dedicated T domains *via* a thioester bond in accordance with a mechanism proposed by Yu et al.[45,46] (Fig. 3b). The condensation domain (C) catalyzes dipeptide formation by nucleophilic attack of the cysteine amino group onto the alanine thioester. This is followed by Cy-domain-mediated thiazoline formation and DH-catalyzed aromatization to the corresponding thiazole[47]. As a second molecular building block, D-tryptophan is decarboxylated by the DC to give tryptamine (**3**), as observed as a product in our expression experiments (Fig. 3a). The terminal amino function of **3** attacks the thioester in the thiazoline intermediate to finally release **4** from the NRPS system.

Bacillamide D (**4**) was initially discovered from *Thermoactinomyces* strain TM-64[48]. Further congeners of the bacillamide family include bacillamides A, B, and C (**5**), which have all been isolated from marine *Bacillus* species[49,50]. In **5**, the terminal amino function of **4** is *N*-acetylated, while in

bacillamide A and B, it is replaced by a keto- or hydroxy function, respectively. Previous biological evaluation of this NP class revealed strong algicidal activity[51], particularly for **4** and bacillamide A, but no activity with direct relevance for human health was reported. Given our discovery of **4** from a bacterial gut isolate, the broad evaluation of its potential effects in an intestinal context was thus of great interest. As the heterologous expression experiment exclusively delivered **4**, as expected from the genetic

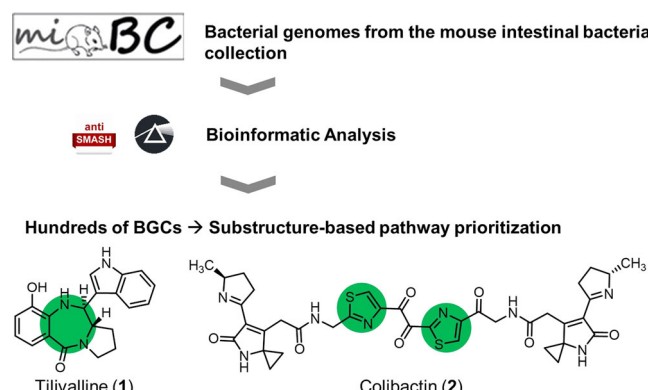

**Fig. 1 | Workflow of this study.** The genomes of isolates within the mouse intestinal bacterial collection (miBC)[72] were bioinformatically screened for de novo biosynthetic potential with antiSMASH[36] and Prism[37] (Logos licensed under CC BY 4.0). Based on the functional importance of heterocyclic structural features of bioactive NPs, such as tilivalline (**1**) and colibactin (**2**), the identified BGCs were prioritized with respect to the presence of heterocyclization enzymology.

composition of the encoding BGC, we additionally prepared a second bacillamide analog, **5**, by semi-synthesis. We selected **5** as an interesting target due to its drastically altered lipophilicity (polarity of a primary amine versus an amide) along with straightforward semi-synthetic access by a simple *N*-acetylation reaction. In addition, *N*-acetylation of **4** might also occur in vivo, e.g. by non-enzymatic condensation with cellular acetyl-CoA. Installation of the *N*-acetyl group was readily achieved using acetic acid anhydride and pyridine as the base in tetrahydrofurane (THF), furnishing **5** in sufficient 31% yield after purification by preparative HPLC (Fig. 4).

## Evaluation of the biological effects of tryptamine, as well as bacillamides 4 and 5

To evaluate the biological activity profile of the bacillamides in the context of the human gut microbiome and human health, we initially started with screening for antibiotic activities. Many bacterial NPs possess antimicrobial properties, granting the producing organism selection advantages in their environments. In the context of the gut microbiome, such activity could be involved in overall gut microbial composition and function. Antibiotic activity of bacillamide D (**4**) was hence initially tested on a panel of pathogenic S2 bacterial strains, including four ESKAPE pathogens: *Staphylococcus aureus* USA300 Lac (JE2), *Klebsiella pneumoniae* subsp. *pneumoniae* DSM 30104, *Acinetobacter baumannii* DSM 30007, *Pseudomonas aeruginosa* PAO1, *Enterobacter cloacae subsp. cloacae* DSM 30054, *Escherichia coli* 536, *Enterococcus faecalis* V583, and *Listeria monocytogenes* EGD-e. No antibiotic effect could be observed in all tested strains at concentrations up to 1 mM (see Table S1), generally making decisive antibiotic effects of **4**, also within the gut microbiome, unlikely. The biological activity profile of tryptamine (**3**) in the context of the gut microbiota has been well-investigated in literature[52–54]. We thus did not evaluate the activity of this *bac*

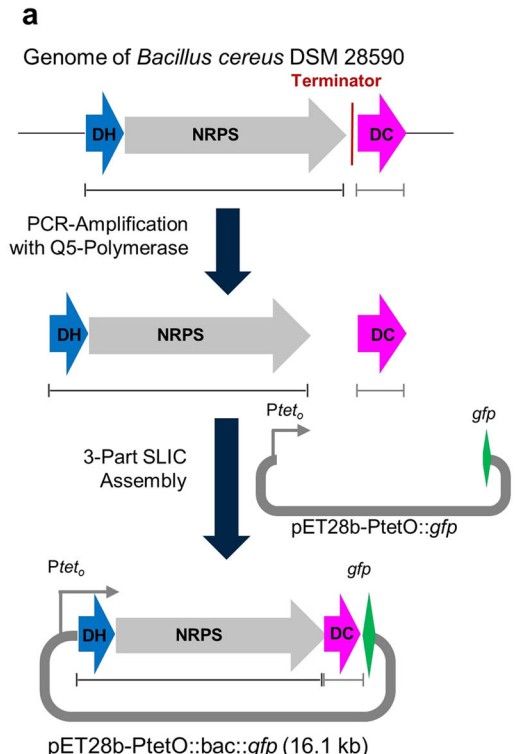

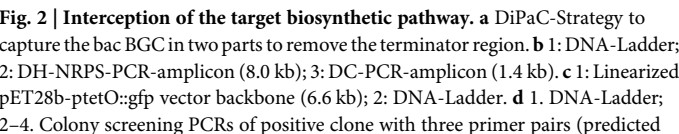

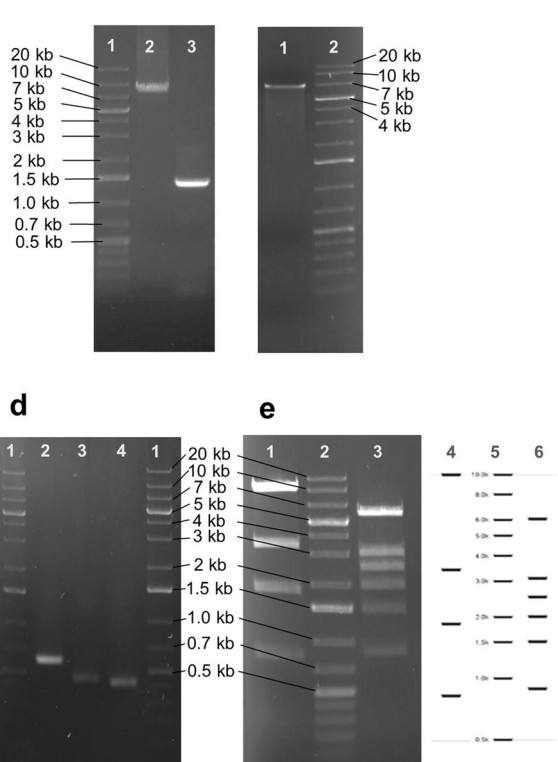

**Fig. 2 | Interception of the target biosynthetic pathway. a** DiPaC-Strategy to capture the bac BGC in two parts to remove the terminator region. **b** 1: DNA-Ladder; 2: DH-NRPS-PCR-amplicon (8.0 kb); 3: DC-PCR-amplicon (1.4 kb). **c** 1: Linearized pET28b-ptetO::gfp vector backbone (6.6 kb); 2: DNA-Ladder. **d** 1. DNA-Ladder; 2–4. Colony screening PCRs of positive clone with three primer pairs (predicted results: 2: 597 bp, 3: 455 bp, 4: 427 bp). **e** 1: Restriction digest of pET28b-ptetO::bac::gfpv2 with EcoRI; 2: DNA-Ladder; 3: Digest with EcoRV; 4: Virtual digest with EcoRI; 5: DNA-Ladder for virtual digest; 6: Virtual digest with EcoRV (virtual digests predicted using Geneious).

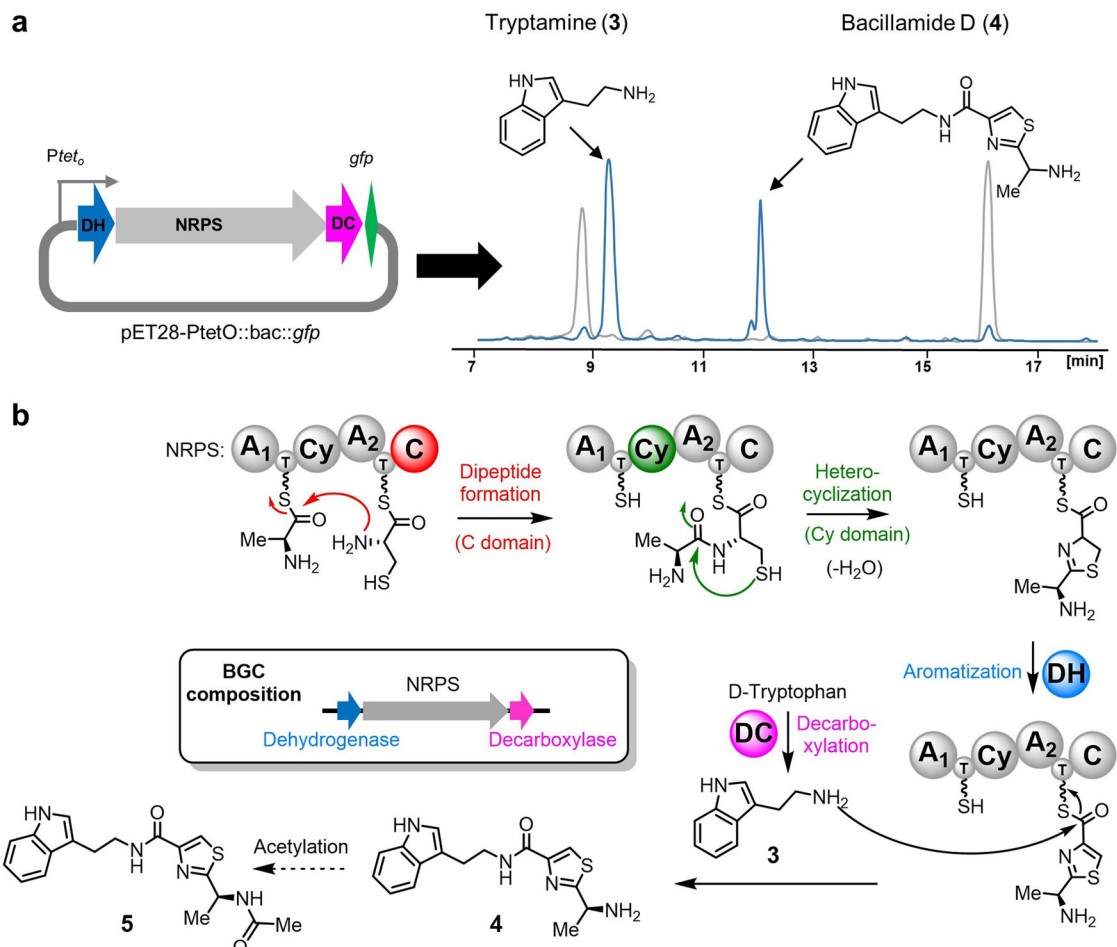

**Fig. 3 | Bacillamide recombinant pathway expression and biosynthesis.**
**a** Heterologous expression of the bac BGC cloned from a B. cereus mouse gut isolate yields tryptamine (**3**) and bacillamide D (**4**). **b** Biosynthesis of the bacillamides.

NRPS non-ribosomal peptide synthetase, A A domain, Cy cyclization domain, C condensation domain, DH dehydrogenase, DC decarboxylase.

pathway intermediate in detail in this study but rather concentrated on the final pathway product **4**.

We next turned our attention to evaluating potential effects on eukaryotic cells. Cell proliferation assays were performed on the human colorectal cancer cell line HCT116 and the embryonic kidney cell line HEK293[55]. Initial viability assays by crystal violet staining in a 96-well plate format revealed the onset of cell growth inhibition of bacillamide D (**4**) between concentrations of 10 µM and 250 µM against both cell lines (Figure S2). Notably, the precursor compound tryptamine (**3**) showed only weak cytotoxic activity, as tested by viability and XTT metabolic growth assays (Figure S3). In-depth evaluation of concentration- and time-dependent activity of **4** using XTT proliferation assays showed highly potent effects, with $IC_{50}$ values of 22.3 µM and 9.70 µM against HEK293 as well as 14.7 µM and 10.2 µM against HCT116, each after 24 and 72 hours, respectively (Fig. 5a). Comparison of the biological effects of **4** and bacillamide C (**5**) demonstrated only slightly decreased activity for the latter, with $IC_{50}$ values of 36.9 µM (HEK293, 24 h) and 23.0 µM (HCT116). Of note, the observed cytotoxicity of both compounds in our tests is thus in the range of clinically established cytotoxic drugs, such as fluorouracil[56] and oxaliplatin[57], both used in treating colorectal cancer. Given this robust activity, we additionally evaluated further essential parameters for self-renewal or carcinogenesis in the gastrointestinal tract, such as cell motility. Translocation is a crucial ability of living cells, and malignant cancer cells especially rely on their migratory ability to invade adjacent tissue, a prerequisite for metastasis formation[58]. Both compounds **4** and **5** reduced the spontaneous 2D migration of HCT116

cells by ~70–80%, depending on the duration of the experiment, essentially without inducing cell death during the time course of the assay (Fig. 5b). As observed before, **4** showed slightly stronger activity than **5**. These findings show that the bacillamides not only possess the ability to suppress the growth of eukaryotic cells but also strongly impede their migratory ability. In addition, we evaluated the potential effects of the more potent **4** on primary cells originating from the murine intestine. We initially selected the duodenal mouse epithelial MODE-K cell line[59] and incubated the cells in 5, 10, 50, 100, 250, and 500 µM final concentration of **4** for 24 h and 72 h. Substantially reduced viability compared to vehicle controls and untreated cells was observed, especially at higher concentrations. Interestingly, incubation after 24 h and 72 h in this case showed only minor differences in the $IC_{50}$ value with 41.2 µM and 27.5 µM, respectively (Fig. 5c).

The putative cell uptake of the more potent bacillamide D (**4**) into live HCT116 cells was investigated by HPLC measurement of fractionated cell extracts. Cells were harvested at different time points (2 h, 24 h, 48 h) after treatment with 100 µM **4** and fractionated into the crude membrane, cytoplasmatic, and nuclear fractions. Aliquots of the samples were tested by immunoblot analysis for the presence of marker proteins for the respective fractions (lamin A/C as nucleus marker, tubulin as marker for the cytoplasm, and β1-integrin for the crude membrane fraction, Figure S4). The fractions were lysed in 100% methanol and measured by HPLC after removal of solid residues by centrifugation. Comparison of the resulting chromatograms with extracts of untreated cells indeed allowed the identification of **4** in the cytosol of the 24 h and 48 h samples, with only traces in

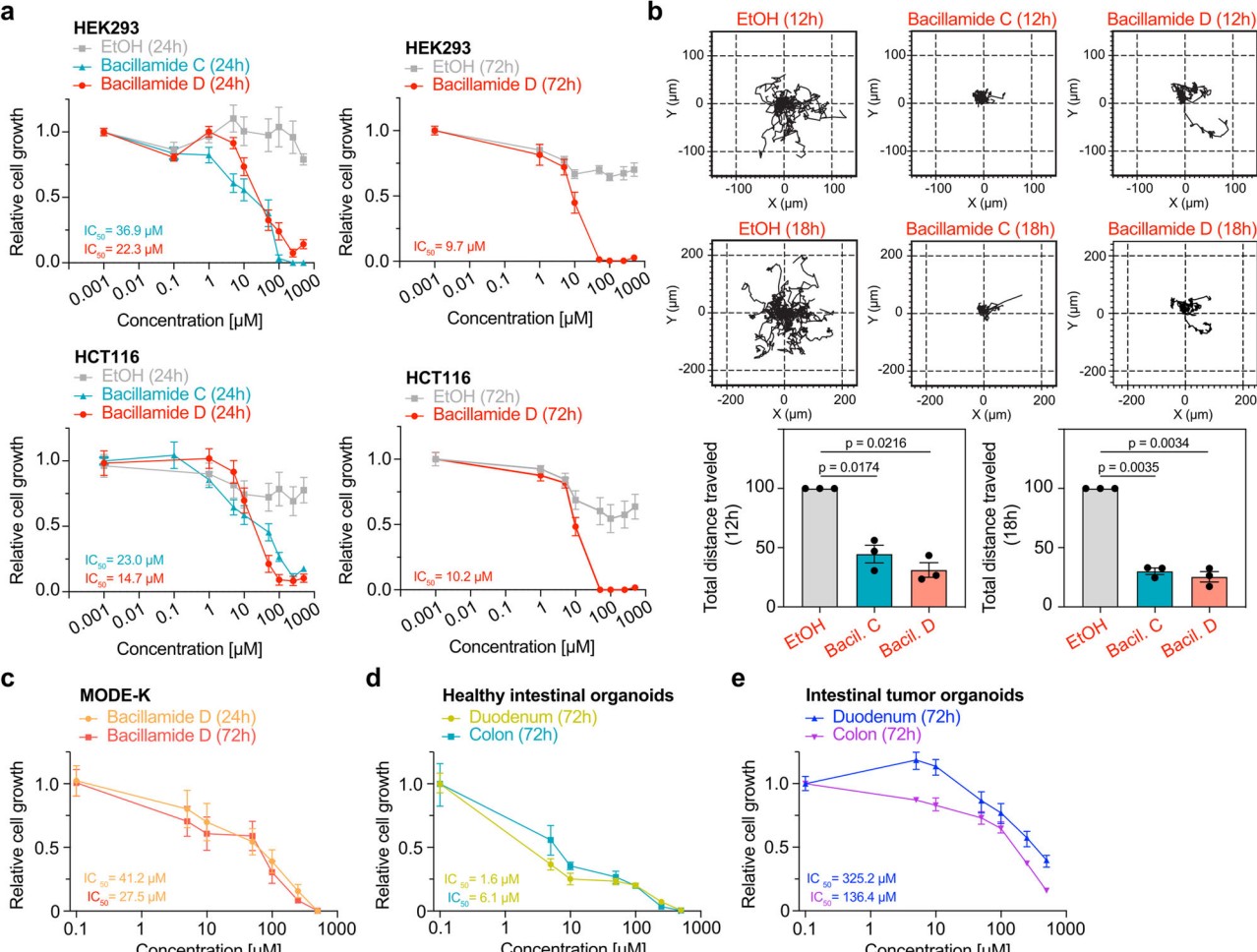

**Fig. 4 | Semi-synthetic preparation of bacillamide C (5).** Heterologously produceed bacillamide D (**4**) undergoes acetylation in THF with acetic acid anhydride as acetylating agent and pyridine as base.

**Fig. 5 | Heterologously produceed bacillamide D (4) undergoes acetylation in THF with acetic acid anhydride as acetylating agent and pyridine as a base. a** XTT proliferation assays were carried out in technical triplicates. The activity of bacillamides C (**4**) and D (**5**) against HEK293 human kidney and HCT116 colon cancer cell lines after 24 h ($n = 6$) and of **4** after 72 h ($n = 3$) is shown with respective IC$_{50}$-values, derived in GraphPad Prism (8.0.2) after log transformation of raw values, followed by non-linear regression analysis of dose-response inhibition. **b** HCT116 cells show substantially decreased motility after 12 h treatment with Bacillamide C ($p = 0.0174$) or Bacillamide D ($p = 0.0216$) compared with vehicle controls, the effect was even more pronounced after 24 h treatment (Bacillamide C, $p = 0.0035$; Bacillamide D, $p = 0.0034$; $n = 3$ assays, mean ± standard deviation is shown, unpaired $t$ test). **c** Growth inhibition of Bacillamide D against murine MODE-K cell line ($n = 4$ experiments, normalized against untreated controls). **d, e** Growth inhibition of Bacillamide D against healthy intestinal organoids (**d**) and intestinal tumor organoids (**e**), each organoid assay was carried out twice with technical triplicates for each assay, normalized against the mean of vehicle-treated controls.

the 2 h incubation sample (Figure S5). No detectable amounts of **4** were found in the crude membrane- and nuclear fractions, clearly indicating its enrichment in the cytosol.

We continued the viability experiments with organoid models to validate the cytotoxic effects of bacillamide D (**4**) in a more complex in vitro system. Therefore, we isolated intestinal stem cells from the duodenum and from the colon of a healthy C57Bl/6 J mouse to generate organoids that recapitulate the cellular heterogeneity and architecture of the intestine. To explore the potential of treatment strategies, we also wanted to assess and compare the viability of intestinal tumor-derived organoids after treatment

with **4**. Two mouse models commonly used in CRC to study carcinogenesis were utilized as donors for tumor organoid isolation. Transgenic mice with *Villin*-promoter-driven tissue-specific recombination in the intestine[60] of either *Kras*$^{G12D}$ [61] or *Braf*$^{V637E}$ [62] were used to generate duodenal and colonic-derived tumors, respectively. For comparability reasons, we chose organoid lines, of which both the original tumors had reached a histopathological score equaling the highly dedifferentiated adenocarcinoma G3 state. These organoids were incubated with the more potent **4** for 72 h under the abovementioned conditions. Intriguingly, **4** affected growth strongest in healthy intestinal organoids (IC$_{50}$ of 1.6 μM to 6.1 μM, Fig. 5c), while the

viability of the tumor-derived organoids was least affected (IC$_{50}$ of 136.4 μM to 325.2 μM, Fig. 5c).

In conclusion, here we introduce our substructure-based pathway prioritization rationale to select promising BGCs from gut bacterial isolates based on their bioinformatically predicted heterocyclic elements. Applying this approach, we discovered bacillamide D (4) from a gut-derived *B. cereus* strain by capture of such a BGC with DiPaC and subsequent heterologous expression in *E. coli*. 4 is established as a highly potent cytotoxin, being selectively taken up into the cytosol of human cells, inhibiting growth with IC$_{50}$-values between 2 and 27 μM across all tested cell lines and reducing cellular motility drastically, with its *N*-acetylated derivative bacillamide C (5) showing similar activity. Similar effects were obtained with healthy intestinal organoids. The cytotoxicity was also observed in assays with murine tumor organoids, albeit not reaching the IC$_{50}$ values obtained with 2D cultures. This may reflect differing uptake kinetics and distribution of the compounds between conventionally cultured 2D cells and 3D organoids growing as multicellular structures in a tumor extracellular matrix protein mixture (Matrigel). In addition, generic tumor cell mechanisms that confer higher resistance against apoptotic cell death and resistance against xenobiotics could also be involved. Taken together, these findings emphasize that the activity of the bacillamides on human and murine cells, in general, can be considered detrimental. Studies on the mode of action of 4 and 5 and their effects in vivo are currently ongoing in our laboratories.

HRMS analysis of lysed cell pellets confirmed the production of bacillamide D (4) by the native producer strain (Figure S6). It is important to note that the *bac* BGC occurs frequently within the *B. cereus* group, being present in about 2/3 of all sequenced *B. cereus* group genomes (see Figure S7). Members of the *B. cereus* group are ubiquitously present in natural environments and have been linked to numerous illnesses. As foodborne pathogens, some *B. cereus* group strains can produce the dodecadepsipeptide cereulide and/or protein-based enterotoxins, allowing them to cause foodborne emetic intoxication and foodborne diarrheal toxic infection, respectively[63–65]. The herein-identified cytotoxic activity of the bacillamides in the context of the human gut emphasizes the need to comprehensively understand the metabolic interplay in the gut in the context of human intestinal health.

## Material and methods
### Strains, plasmids, cell lines, DNA extraction
Bacterial strains and plasmids used in this study are listed in Table S2. *B. cereus* DSM 28590 was cultured in LB medium at 37 °C. *E. coli* strains were grown in LB medium supplemented with 50 μg/mL kanamycin at 37 °C unless otherwise specified. Liquid cultures were incubated while shaking at 180 rpm. Human colorectal cancer cell line HCT116 (RRID: CVCL_0291), and embryonic kidney cell line HEK293 (CVCL_0045) were obtained directly from American Type Culture Collection (ATCC, Rockville, MD, USA) and cultured in DMEM (Invitrogen, Karlsruhe, Germany) containing 7% Fetal Bovine Serum (Biochrome, Berlin, Germany) supplemented with 1% penicillin, 1% streptomycin, 1% L-glutamine (Invitrogen). The SV40 large T-antigen–immortalized murine small IEC line MODE-K (gift from D. Kaiserlian, Institute Pasteur, Paris, France) was maintained in DMEM supplemented with 10% fetal bovine serum, 1% L-glutamine, non-essential amino acids, sodium pyruvate, 1% penicillin, and 1% streptomycin[59]. All cells were tested for mycoplasma infection by PCR at least every 6 weeks, and all experiments were performed with mycoplasma-free cells. To avoid contamination and phenotypic changes, cells were kept as frozen stocks and cultured consecutively for 4 weeks maximum, as published earlier[55].

Extraction of high-molecular-weight genomic DNA from *B. cereus* DSM 28590 for BGC cloning was performed following a pre-established protocol with few adjustments[38,40]. After cultivation, cells were centrifuged and washed with 0.9% NaCl. Cell pellets were resuspended in 500 μL lysis buffer (25 mM EDTA, 0.3 M sucrose, 25 mM Tris-HCl [pH 7.5])) and underwent three freeze–thaw-cycles (liquid nitrogen/50 °C). Lysozyme (Sigma-Aldrich) was added to a final concentration of 1 mg/mL, and, after incubation at 37 °C for 30 min, RNase (Carl Roth, Germany) was added to a

final concentration of 10 μg/mL, followed by incubation at 37 °C for 60 min. SDS was added to a final concentration of 1% (v/v), and the mixture was incubated at 37 °C for 30 min. After addition of proteinase K (Amresco, USA) to a final concentration of 0.5 mg/mL and incubation at 55 °C for 30 min, NaCl was added to a final concentration of 1 M and cetyl-trimethylammoniumbromide (CTAB) [10% (w/v) in 0.7 M NaCl, Sigma-Aldrich) to a final concentration of 1% (v/v). The cell lysis mixture was incubated for 10 min at 65 °C, and 1 vol of chloroform:isoamyl alcohol (24:1) was added, mixed by inversion, and incubated on ice while gently shaking. After centrifugation at 4 °C, the aqueous phase was harvested and extracted twice with phenol:chloroform:isoamyl alcohol (25:24:1) before addition of 0.6 vol of isopropanol. The precipitated DNA pellet was washed twice with ice-cold 70% ethanol, dried at 60 °C to completely remove ethanol residues and finally re-dissolved in TE buffer (10 mM Tris-HCl, 1 mM EDTA [pH 8.0]). The quality of gDNA was analyzed by gel electrophoresis using 0.7% (w/v) agarose gels and PCR with primers specific to the 16S rRNA region of the bacterial genome, which produced a 1.5 kb PCR product. The gDNA was quantified with a P330 NanoPhotometer (Implen, Germany) and stored at −20 °C until use.

### Bioinformatics
The sequenced genomes of miBC strains were downloaded from the NCBI database. Analysis of BGCs was performed using AntiSMASH (Version 5)[35] and PRISM (Version 3)[66]. A total of 100 genomes were analyzed, which led to the identification of 114 BGCs across all major known natural product families (Figure S9). These BGCs were screened using the AntiSMASH program output and manual curation by BLAST for the presence of genes encoding proteins that might be involved in the production of heterocyclic NPs, with a focus on small- to midsized clusters (<40 kb). This led to a shortlist of four NRPS-BGCs from *Staphylococcus xylosis* and *B. cereus*. The bac-BGC was of particular interest because of its small size and simple architecture, greatly facilitating cloning, and was therefore selected for this study. Analysis of the other candidate BGCs is currently ongoing in our laboratory. To visualize nucleotide sequences and design primers, the Geneious software package (Version 8.1.9) was used[67]. Transcriptional terminators within the *bac* gene cluster were predicted by ARNold (http://rssf.i2bc.paris-saclay.fr/toolbox/arnold/)[41].

### Direct pathway cloning
Oligonucleotides used for PCR reactions in this study are listed in Table S3. The 9.5 kb bacillamide BGC of *B. cereus* DSM 28590 was cloned in two inserts into pET28b-ptetO-*gfp* utilizing a one-step three-part SLIC reaction[68]. Linear DNA-fragments of inserts and backbone (Fig. 2c) were generated with PCR using Q5-polymerase (NEB) in 25 μL batches consisting of: 1 × Q5 reaction buffer, 200 μM deoxynucleotide triphosphates, 500 nM of forward and reverse primer, 50 ng gDNA template and 0.01 U/ μL Q5 High-Fidelity DNA polymerase (NEB). Thermal cycling was performed in a T100 Thermal Cycler (Bio-Rad) as follows: (1) Initial denaturation, 98 °C for 30 sec.; (2.) Denaturation, 98 °C for 10 sec.; (3) Primer annealing for 20 sec.; (4) Extension, 72 °C for 45 s/kb; (5) Final extension, 72 °C for 5 min. Steps (2) to (4) were repeated for 30 cycles in total. The annealing temperatures for specific primer pairs were estimated with the NEB Tm Calculator tool (https://tmcalculator.neb.com/). The reaction mixture from the backbone linearization reaction was digested with DpnI prior to purification to remove remaining circular template plasmid.

At the 5′ end of the insert-specific primer pairs, 23–33 bp homology sequences consistent with the terminal region of the linearized backbone or the other insert were added (Table S3). After agarose gel purification, the fragments were assembled in a 10 μL SLIC reaction with T4-DNA polymerase and Buffer 2.1 (both NEB). The reaction was incubated at room temperature for 150 sec. and kept on ice for 10 min before the transformation of 5 μL into *E. coli* DH5α by heat shock. The cells were plated on LB-agar with kanamycin as a selection antibiotic, and transformants were initially screened using colony-PCRs designed to amplify overlapping regions at the fusion sites in the assembled expression plasmid (Fig. 2d). For

colony-PCR using Onetaq Polymerase (NEB), clones were picked, resuspended in 12 μL of LB medium and examined in a 25 μL PCR reaction composed as following: $1 \times$ Onetaq Buffer, 200 μM deoxynucleotide triphosphates, 200 nM of forward and reverse primer, 2 μL cell suspension (DNA template) and *Taq* DNA polymerase (NEB). Thermocycling conditions were set as described above, with the exception of 94 °C denaturation and 68 °C extension temperatures. One positive clone underwent restriction digest (Fig. 2e) and whole plasmid sequencing (Figure S1, https://www.plasmidsaurus.com/), confirming successful cloning.

### Heterologous expression of the bacillamide gene cluster
Culture conditions for heterologous expression were based on previously described experiments for the pET28b-ptetO plasmid[38,39]. Prior to each expression experiment, *E. coli* BAP1 was chemically transformed with pET28b-ptetO::*bac*::*gfp* and pET28b-ptetO::*gfp* (empty) as negative control. Precultures were grown overnight from a single clone and used to inoculate expression cultures (1% v/v) of 1 L TB medium in 2 L Erlenmeyer flasks, supplemented with 50 μg/mL kanamycin. Expression cultures were incubated while shaking at 37 °C for 150–180 min until reaching an $OD_{600}$ of 0.8 and then cooled to 10 °C. Expression was induced by adding 0.5 μg/mL tetracycline, and the cultures were incubated for 5 days at 20 °C in darkness to avoid light-induced decomposition of tetracycline.

### Extraction and isolation
The expression cultures were centrifuged ($6000 \times g$ for 20 min) to separate *E. coli* biomass from growth medium. Cell pellets were extracted with methanol (50 mL per 1 L culture) by incubation in a sonicator bath for 15 min. Cell debris was removed by centrifugation, and the solvent was removed under reduced pressure at 40 °C. Culture supernatants were adjusted to a pH of 9.0 with saturated $NaHCO_3$-solution and extracted with ethyl acetate ($2 \times 0.7$ L per liter of growth medium). The combined organic phase was washed with 0.1 L of saturated brine and dried over $MgSO_4$, and the solvent was removed under reduced pressure. Dried extracts were dissolved in HPLC-grade methanol and filtered through a syringe-driven 0.2 μm PTFE membrane filter (Fisherbrand, USA) prior to HPLC analysis.

Analytical HPLC experiments were carried out on a Knauer Azura HPLC system, consisting of the following components: AS 6.1 L sampler, P 6.1 L pump, DAD 2.1 L detector. The chromatographic HPLC separation was conducted on a Knauer Eurospher II 100-3 C18 column ($150 \times 4.6$ mm, 3 μm particle size). Water (A) and acetonitrile (B) were used as eluents, both supplemented with 0.05% trifluoroacetic acid. Samples were eluted at 1 mL/min using the following gradient: 5% B for 2 min followed by a gradient to 100% B over 28 min. Column washing was performed at 100% B for 5 min, and the column was re-equilibrated at 5% B for 2 min before the next measurement.

Compounds were purified on a preparative Jasco HPLC system consisting of an UV-1575 Intelligent UV/Vis detector, two PU-2068 Intelligent preparation pumps, a Mika 1000 dynamic mixing chamber (1000 μl; Portmann Instruments AG Biel-Benken) and a LC-NetII/ADC anda Rhodyne injection valve with a Knauer Eurospher II 100-5 C18 Column ($250 \times 16$ mm, 5 μm particle size). Eluents were used as described above with a modified gradient of 15–40% B over 15 min, followed by column washing at 100% B for 3 min and re-equilibration at 15% B for 2 min, at a flow-rate of 10 mL/min. Tryptamine (**3**) and bacillamide D (**4**) were eluted at 6.5 and 9.5 min, respectively, and isolated as slightly yellow oils (yields: tryptamine (**3**): 20 mg/L; bacillamide D (**4**): 77 mg/L).

### NMR-measurement
$^1$H and $^{13}$C Nuclear Magnetic Resonance spectra (NMR) were recorded on Bruker AVANCE 300 and AVANCE 600 spectrometers at room temperature. The chemical shifts are given in δ-values (ppm) downfield from TMS and are referenced on the residual peak of the deuterated solvent (DMSO-d6: $\delta_H = 2.50$ ppm, $\delta_C = 39.5$ ppm; $CDCl_3$: $\delta_H = 7.26$ ppm, $\delta_C = 77.0$ ppm; Methanol-d4: $\delta_H = 3.31$ ppm, $\delta_C = 49.0$ ppm). The coupling constants *J* are given in Hertz [Hz]. Following abbreviations were used for the allocation of signal multiplicities: s—singlet, d—doublet, t—triplet, *virt*. t —virtual triplet, q—quartet, *virt*. q—virtual quartet, m—multiplet.

### Specific rotation
Specific rotations were measured with a Krüss P3000 polarimeter at 20 °C in methanol.

### HRMS-measurement
HRMS was performed on a Bruker Impact II ultra-high-resolution Q-TOF mass spectrometer with electron-spray ionization (ESI).

### Minimal inhibitory concentration measurements
Minimal inhibitory concentrations of bacillamide D against a panel of difficult-to-treat pathogens (Table S1) were determined in a 96-well plate format (transparent Nunc 96-well flat bottom, Thermo Fisher Scientific). In general, the wells of the edges from each plate were deposited with 100 μl medium to prevent evaporation effects. A stock solution of bacillamide D at a concentration of 100 mM in DMSO was prepared for subsequent serial dilutions: 100 μl of twofold the highest concentration to be tested (1 mM) was deposited in the wells of the second column. The remaining wells were deposited with each 50 μl of respective medium (Table S1) supplemented with 2% DMSO. By using a multichannel pipette, 50 μl of the second column was transferred to the third column by thoroughly mixing, and so on, with the exception of the last column, which was used as a growth control without bacillamide D.

Bacterial cell suspensions were prepared in 1:1000 dilutions from o/n cultures in respective mediums, ensuring a stationary phase. 50 μl of each bacterial cell suspension was added to the wells in duplicates with the deposited bacillamide D solutions and the column for the growth control. Microtiter plates were incubated for 24 h (37 °C, 200 rpm), and serial dilutions were analyzed for microbial growth, indicated by turbidity. MIC values of bacillamide D were considered to be the lowest compound concentrations where no bacterial growth was observed by eye.

### Cell proliferation assays
Cell proliferation in human cell lines was determined by mitochondrial metabolic activity using a Cell Proliferation Kit II (XTT) (Roche, Mannheim, Germany), essentially as described[55]. In all, $3 \times 10^3$ cells were seeded to each well of 96-well plate and incubated overnight, allowing cells to adhere. Cells were then treated with indicated concentrations of bacillamide C (**5**) or D (**4**) (both dissolved in ethanol), for 24 h or 72 h. As solvent control, ethanol was added at the volume corresponding to the highest concentration of bacillamides. Cell proliferation was then measured according to manufacturer's instruction. $IC_{50}$ values were calculated with GraphPad Prism 8.0.2.

### Cell survival and cell migration assay
Two independent methods were implemented to assess cell survival assay. Briefly, survival of adherent cells was tested with a crystal violet staining assay. $1 \times 10^5$ cells were seeded to each well of a six-well plate and incubated for 24 h before treatment to allow for cell adhesion. Next, 10 μM and 250 μM of bacillamide D (**4**) or equal amounts of alcohol (vehicle control) were added to the wells and then incubated for further 72 h. Cells were then fixed in 3% paraformaldehyde for 5 min, washed briefly, and stained with 0.1% Crystal Violet solution (Sigma, St. Louis, MO, USA) for 30 minutes to visualize adherent cells. The dye was then extracted from fixed cells by adding 2 ml methanol to each well and incubation for 20 min on a shaker at room temperature. The extracted dye was then measured at 540 nm absorbance in a microplate reader (Berthold, Bad Wildbad, Germany), essentially as published earlier[69]. As published recently[7], live cell imaging was used as a second method to directly monitor cell survival, expressed as relative cell coverage per field of view, during treatment. Briefly, cells were seeded at $3 \times 10^4$ cells per well (24-well plate) and incubated overnight at 37 °C. Cells were then treated with 10 μM or 250 μM of **4** and **5** or equal amounts of vehicle control, and monitored on a 24-channel incubator

microscope by phase contrast (zenCellOwl, innoME GmbH, Germany) in a cell culture incubator (37 °C, 7% CO$_2$, 95% humidity) for 60 hours. Pictures were taken automatically every 15 minutes, or 10 minutes for cell migration assays. The same setup was used to monitor spontaneous cell migration in 2D with ImageJ Fiji software (Version 1.0, based on the plugin Manual-Tracking; NIH, Bethesda, MD, USA).

## Cell fractionation

Cell fractionation was performed following a modified protocol, based on the protocol described by Franke et al. [56] Per well, $4 \times 10^6$ cells were seeded, allowed to attach overnight, and incubated for indicated time points with the compounds **4** and **5**. Cells were washed once in cold PBS, lysed on the plate with 600 μL hypotonic cell lysis buffer (10 mM HEPES pH 7.4, 10 mM NaCl, 5 mM NaHCO$_3$, CaCl$_2$ 1 mM, MgCl$_2$ 0.5 mM, 5 mM EDTA, 0.1% NP-40, 1X Complete EDTA-free Protease Inhibitor Cocktail (Roche 4693132001), 1 mM Pefabloc), and harvested by scraping. This harvested cell lysate was then incubated for 10 min on ice, homogenized manually with a glass Dounce homogenizer 50×, and then transferred to a 1.5 mL centrifuge tube. The dounced lysate was centrifuged (4000 rpm for 5 min at 4 °C). The resulting suspension (S1) was processed to yield whole cell lysate, crude cytosolic fraction, and crude membrane fraction, whereas the resulting pellet (P1) was processed to yield the nuclear fraction.

For whole cell lysates, 200 μL aliquot of S1 supernatant was transferred to a clean microcentrifuge tube, mixed with 50 μL methanol, sonicated on ice (2.5 sec. intervals for 10 min; Bioruptor, Biogenode, Belgium), centrifuged (15,000 rpm for 10 min at 4 °C) to remove remaining debris, and the final supernatant was used whole cell lysate for further analysis, e.g. by HPLC or immunoblot. To yield crude cytoplasmic and membrane fractions, respectively, another 300 μL aliquot of S1 was centrifuged (15,000 rpm for 15 min at 4 °C). 200 μL of the resulting supernatant (S2) was mixed with 50 μL methanol, sonicated on ice as detailed above, centrifuged (15,000 rpm for 10 min at 4 °C). The resulting supernatant (S3) was used as crude cytoplasmic fraction, and the pellet (P3) discarded. The rest of S1 was discarded, and the pellet (P2) was further used to isolate the crude membrane fraction by resuspending it in a mix of 200 μL ddH2O and 50 μL methanol. The mixture was sonicated on ice as described above and centrifuged for 10 min at 15,000 rpm at 4 °C. The resulting supernatant (S4) was then transferred to a new tube and used as crude membrane fraction for further analysis.

To isolate the nuclear fraction, P1 was resuspended in 800 μL TSE-Buffer (10 mM Tris pH 7.5, 300 mM Saccharose, 1 mM EDTA, 0.1% NP-40), homogenized manually with a glass Dounce homogenizer 30 times, and centrifuged (4000 rpm for 5 min at 4 °C). While the resulting supernatant was discarded, the nuclei-containing pellet was further processed by washing it in 800 μL TSE buffer and centrifugation (3000 rpm for 5 min at 4 °C) to remove remaining contaminations. The supernatant was discarded, and the nuclei pellet was finally resuspended in 200 μL RIPA buffer (50 mM Tris pH 7.4, 1% NP-40, 0.25% Na-deoxycholate, 150 mM NaCl, 1 mM EDTA, 1X Complete EDTA-free Protease Inhibitor Cocktail (Roche 4693132001), 1 mM Pefabloc, 0.1% NP-40). 50 μL methanol was added to the resuspended pellet, sonicated on ice as above, and centrifuged (15,000 rpm for 10 min at 4 °C). This final supernatant was transferred to a clean microcentrifuge tube and used as nuclear fraction for further analysis.

## Organoid isolation and cultivation

Organoids from the duodenum and colon were generated as previously described[70]. Briefly, intestines were rinsed with ice-cold PBS and cut into ~5 mm pieces. These pieces were incubated for 30 min in a 5 mM EDTA (Sigma-Aldrich) solution in PBS for small intestine and 30 mM EDTA for colon organoids and intermitted by gentle shaking every 10 min followed by a vigorous shaking in ice-cold PBS for 3 cycles of 30 sec. Cell suspension was filtered through a 100 μm and a 70 μm cell strainer (Greiner Bio-One) and spun down at $300 \times g$ for 5 min. Around 50–100 crypts were resuspended in 50 μL of Matrigel (Corning) and plated in a well of a 24-well plate (Corning). To generate tumor-derived organoids, tumors were minced with a scalpel

and incubated with 1% PenStrep (Thermo Fisher Scientific) in PBS for 15 min on ice. After washing, the tissue was digested with 230 U of collagenase type IV (Merck) at 37 °C for 30 min, filtered through a 100 μm strainer, and washed with PBS followed by a 10 min inactivation of the collagenase by incubation with 10% FCS for 10 min. After centrifugation at $300 \times g$ for 5 min ~100 cell clumps were resuspended in 50 μL of Matrigel and plated in a well of a 24-well plate. A conditioned medium (50% diluted) containing Wnt3A, R-spondin, and Noggin, produced by L-WRN cells (ATCC® CRL3276™), was supplemented with 10 μM of Y-27632 (STEM-CELL Technologies) and added to the organoid cultures. Medium was changed every 2–3 days, and the culture passaged every 4–6 days. Passaging of organoid cultures was done by incubation and mechanical disassociation of the culture with Cell Recovery Solution (Corning) for 30 min on ice, followed by centrifugation step at $300 \times g$ for 5 min. Cells were enzymatically digested by an incubation with TypLE Express Enzyme (Thermo Fisher Scientific) diluted to 1:2 for normal tissue and 1:1 for tumor tissue organoids in PBS and incubated at 37 °C for 2–3 min. Cells were then washed once with PBS, centrifuged at $300 \times g$, resuspended in Matrigel, and again plated in a 24-well plate (50 μL/well).

## Cell viability assay

In 2D cultures, 5000 cells in 150 μL of media were seeded in triplicates or quadruplicates per treatment concentration and per timepoint in a white 96-well flat bottom plate (Corning). After 24 h of attachment, another 150 μL containing bacillamide D (**4**) for final concentrations of 5, 10, 50, 100, 250, and 500 μM in ethanol or only vehicle in media was added. For 3D cultures, 15,000–20,000 cells were seeded in 10 μL of a 10% Matrigel in PBS mixture in triplicates or quadruplicates per treatment concentration in a white 96-well flat bottom plate (Corning). The compound diluted in ethanol or only vehicle was diluted in 50% L-WRN (ATCC® CRL3276™) conditioned medium to a total of 150 μL, and organoids were allowed to grow for 72 h. To assess cell viability on the specific timepoint, the CellTiter-Glo® (CTG) Luminescent Cell Viability Assay (Promega, Cat# G7573) reagent was equalized at room temperature for ~30 min previous to usage. The supernatant was partly removed, and CTG reagent was added 1:1. In organoid cultures, well content was vigorously mixed to homogenize the solution with Matrigel and to induce cell lysis. After incubation for 45 min on a shaker (100 rpm) absorbance was measured at a ClarioStar machine. Values were normalized to their respective untreated control, and IC$_{50}$ values (non-linear regression model) were calculated using GraphPad Prism 8 (GraphPad Software, San Diego, USA).

## Synthesis of bacillamide C (5)

Bacillamide D (**4**, 13.7 mg, 0.0434 mmol, 1 Eq.) was dissolved in 2 mL dry THF, and the solution was cooled to 0 °C. Freshly distilled acetic acid anhydride (4.55 μL, 4.92 mg, 0.0482 mmol, 1.1 Eq.) was added under stirring followed by pyridine (10.5 μL, 10.3 mg, 0.130 mmol, 3.0 Eq.). The reaction was stirred at 0 °C for 30 min followed by 2 h at room temperature. Analytical HPLC indicated over 90% turnover at this point. Finally, the solvent was removed under reduced pressure, and the crude product was purified by preparative HPLC (20–40% B gradient over 15 min, elution at 11 min) to yield **5** as a slightly yellow oil (4.80 mg, 0.0135 mmol, 31%).

## Analytical data

**Tryptamine (3)**. $^1$H-NMR (500 MHz, DMSO-d6) δ 11.01 (s, 1H), 7.97 (s, 3H), 7.56 (d, $J$ = 7.9 Hz, 1H), 7.38 (d, $J$ = 8.1 Hz, 1H), 7.24 (d, $J$ = 2.0 Hz, 1H), 7.10 (*virt* t, $J$ = 7.5 Hz, 1H), 7.01 (*virt* t, $J$ = 7.1 Hz), 3.14—3.05 (m, 2H), 3.0 (t, $J$ = 7.1 Hz, 2H) ppm. $^{13}$C NMR (100 MHz, DMSO-d6) δ 136.4, 126.8, 123.4, 121.2, 118.5, 118.0, 111.6, 109.4, 39.4, 23.2 ppm. HRMS (m/z): [M]$^-$ calcd. for C$_{10}$H$_{11}$N$_2^-$, 159.0928; found, 159.0929.

The spectroscopic data were in agreement with those of commercially available **3** (Acros Organics, China).

**Bacillamide D (4)**. $[\alpha]_D^{20}$ = − 14.7 (29.2 g/L in MeOH) $^1$H-NMR (500 MHz, DMSO-d6) δ 10.86 (s, 1H), 8.74 (s, 3H), 8.49 (t, $J$ = 6.0 Hz,

1H), 8.34 (s, 1H), 7.60 (d, $J$ = 7.9 Hz, 1H), 7.35 (d, $J$ = 8.1 Hz, 1H), 7.19 (d, $J$ = 1.8 Hz, 1H), 7.07 (*virt.* t, $J$ = 7.5 Hz, 1H), 6.98 (*virt.* t, $J$ = 7.4 Hz, 1H), 4.88 (q, $J$ = 7.1 Hz, 1H), 3.60 (*virt.* q, $J$ = 7.6 Hz, 2H), 2.96 (t, $J$ = 7.6 Hz, 2H), 1.64 (d, $J$ = 6.8 Hz, 3H) ppm. $^{13}$C NMR (100 MHz, DMSO-d6) δ 167.5, 160.1, 149.2, 136.3, 127.2, 125.1, 122.6, 121.0, 118.30, 118.28, 111.6, 111.4, 47.6, 25.4, 19.5 ppm. HRMS (m/z): [M]$^+$ calcd. for $C_{16}H_{19}N_4OS^+$ 315.1274; found 315.1278.

The spectroscopic data was in agreement to those described in the literature[71].

**Bacillamide C (5).** $[\alpha]_D^{20}$ = − 40.2 (10.2 g/L in MeOH); $^1$H-NMR (300 MHz, Methanol-d4) δ 8.04 (s, 1H), 7.66 (d, $J$ = 7.8 Hz, 1H), 7.54–7.46 (m, 1H), 7.34 (d, $J$ = 7.4 Hz, 1H), 7.22 (*virt* t, $J$ = 7.0 Hz, 1H), 7.13 (*virt* t, $J$ = 7.7 Hz, 1H), 7.10 (s, 1H), 6.08 (d, $J$ = 7.3 Hz, 1H), 5.42–5.31 (m, 1H), 3.85–3.73 (m, 2H), 3.10 (t, $J$ = 6.8 Hz, 2H), 2.03 (s, 3H), 1.57 (d, $J$ = 6.9 Hz, 3H) ppm. $^{13}$C NMR (75 MHz, CDCl$_3$) δ 172.5, 169.8, 160.9, 149.7, 136.5, 127.6, 123.4, 122.36, 122.34, 119.6, 119.0, 113.2, 111.4, 47.1, 40.0, 25.5, 23.3, 21.6 ppm. HRMS (m/z): [M]$^+$ calcd. for $C_{18}H_{21}N_4O_2S^+$, 357.1380; found, 357.1382.

The spectroscopic data was in agreement with those described in the literature[50].

### Reporting summary
Further information on research design is available in the Nature Portfolio Reporting Summary linked to this article.

### Data availability
All data generated during this study are deposited in the supplementary information and Supplementary Data 1 and 2 and are additionally available from the corresponding authors on request. The expression plasmid pET28b-ptetO::*bac*::*gfp* is deposited in Addgene (ID 217882). Uncropped gel images for Fig. 2b–e and blot images for Figure S4 can be found in the supplementary information (Figures S9-14), as well as $^1$H and $^{13}$C NMR-spectra for compounds 3–5 (Figures S15–17).

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

## Acknowledgements

This work was funded by the Deutsche Forschungsgemeinschaft (DFG, German Research Foundation, project ID 395357507—SFB 1371, Microbiome Signatures to T.C., S.A.S., G.Z., M.T., K.P.J. and T.A.M.G.), SPP 2002 (Small Proteins, an Unexplored World), and INST 269/971-1.

## Author contributions

M.H., T.L., and T.A.M.G. designed the research project. M.H. and T.L. conducted all work associated with BGC cloning, compound expression and isolation, synthesis, and chemical analytics. V.B., W.J., D.S., M.T., and K.P.J. planned and conducted all work on the in-depth characterization of the cyto-toxic effects of bacillamides. D.S. and S.A.S. screened for antibiotic effects.

J.B. and T.C. planned and conducted fermentation of the original producer strain. L.M.C. and G.Z. performed bioinformatic analyses. T.C., S.A.S., G.Z., M.T., K.P.J., and T.A.M.G. provided materials and infrastructure and secured funding for the project. M.H., M.T., K.P.J., and T.A.M.G. wrote the manuscript, which all authors reviewed and revised.

## Funding

## Competing interests

The authors declare no competing interest.

## Additional information

[1]Chair of Technical Biochemistry, Technical University of Dresden, Bergstraße 66, 01069 Dresden, Germany. [2]Chair of Translational Cancer Research and Institute of Experimental Cancer Therapy, Klinikum rechts der Isar, School of Medicine and Health, Technical University of Munich, 81675 Munich, Germany. [3]Center for Translational Cancer Research (TranslaTUM), Klinikum rechts der Isar, School of Medicine and Health, Technical University of Munich, 81675 Munich, Germany. [4]Division of Translational Cancer Research German Cancer Research Center (DKFZ) and German Cancer Consortium (DKTK), 69120 Heidelberg, Germany. [5]Department of Surgery, Klinikum rechts der Isar, School of Medicine and Health, Technical University of Munich, 81675 Munich, Germany. [6]Department of Bioscience, Center for Functional Protein Assemblies, Technical University of Munich, 85748 Garching bei München, Germany. [7]Structural and Computational Biology Unit, European Molecular Biology Laboratory, 61997 Heidelberg, Germany. [8]Research and Development Headquarters, Nitto Boseki Co., Ltd., 102-8489 Tokyo, Japan. [9]Functional Microbiome Research Group, Institute of Medical Microbiology, University Hospital of RWTH Aachen, 52074 Aachen, Germany. [10]Helmholtz Institute for Pharmaceutical Research Saarland (HIPS), Department of Natural Product Biotechnology, Helmholtz Centre for Infection Research (HZI) and Department of Pharmacy at Saarland University, Campus E8.1, 66123 Saarbrücken, Germany. [11]These authors jointly supervised this work: Markus Tschurtschenthaler, Klaus-Peter Janßen, Tobias A. M. Gulder. ✉ markus.tschurtschenthaler@tum.de; klaus-peter.janssen@tum.de; tobias.gulder@tu-dresden.de

