## [Peer Review File · Communications Biology]

Reviewers' comments:

Reviewer #1 (Remarks to the Author):

In this study from Hohmann et al. titled "Bacillamides produced by *Bacillus cereus* from the mouse intestinal bacterial collection (miBC) are potent cytotoxins" the authors first identify a non-ribosomal peptide synthetase gene cluster in the genome of *Bacillus cereus* isolated from mouse fecal sample. The authors proceed to clone genes from the cluster and to identify 2 products: tryptamine and bacillamide D. They further show that the bacillamide D has cytotoxic activity on several cell lines and mouse tumor organoids.

The potential effect of different microbiome-derived compounds on the host's health has been getting a lot of attention lately. Identification of one such molecule that shows cytotoxic effect in tumor organoids is relevant for the field. While in general the authors' conclusions so far are supported by the data they provide, and are interesting, I believe addition of *in vivo* experiments would significantly strengthen the study. Complete list of my comments is listed below:

Major points

1. The results on cytotoxic effect of bacillamide D on tumor organoids are promising, however, the study would greatly benefit from *in vivo* experiments (e.g. administration of bacillamide D to mice with either colorectal cancer or any other tumor model the authors consider appropriate). If the effect persists, it would reinforce the conclusions and increase the relevance of the results. Without these experiments, the study lacks strength.
2. Line 115 and 326: The manuscript is missing information on how the bac was identified and the analysis itself: a) What were the parameters of the analysis? b) What was the output of initial search, or in other words was bac the only BGC that could be identified? It is hard to understand how the authors mined the data and how it led to bac being selected for further analysis. Would this workflow be something that others should explore, or is the identification of bac just one lucky event?
3. Line 185: It is unclear why the authors focus on the bacillamide C and not A or B. The authors state that they chose bacillamide C because of its altered lipophilicity. However, are there previous studies to support this choice? In fact, authors mention that bacillamide A was previously reported to have algicidal activity, together with the bacillamide D they identified, which would make one think the authors would test the bacillamide A as well, but this is not the case.
4. Line 283: Could bacillamide D be detected in mouse or human gut and at what concentrations? It would be beneficial for the study and its relevance if the authors provided some information on either bac or bacillamide D presence in the gut microbiota. Could the authors analyze publicly available data (mouse and human) for the presence of bac? How prevalent and abundant it is? Is it possible that the bac product (bacillamide D) is produced in concentrations that would exert effect on the host?
5. In general, the numbers under the Figure 4 and in method section are missing. What is the number of replicates per group for each of the plots? How many times was each experiment performed? What was the statistical test the authors performed and what are the p values? All this information should be included.

Minor points:

1. The title of the study is somewhat misleading and should be re-phrased: the authors tested only bacillamide D produced by *B. cereus* from miBC. The other bacillamide (C) was the one identified in marine and not murine strain. This is the reason why the title shouldn't say "Bacillamides produced by *Bacillus cereus* from the mouse intestinal bacterial collection (miBC) are potent cytotoxins". Alternatively, authors should show that *B. cereus* from miBC produces bacillamide C as well.
2. Unless the study is completed with mouse experiments, the title should reflect that the results are obtained through experiments performed *in vitro*.
3. Line 36: Missing full stop at the end of the sentence.
4. Line 57: This sentence implies that the NPs are somehow produced by interaction of different members of the microbiome. I would suggest saying "However, *de-novo*-produced NPs from different

members of complex microbial ecosystems, such as the human gut microbiome, and their potential effects on host cells have remained poorly understood”.

5. Lines 59-62: Missing references. Are there references to support these claims or give an example? If not, it seems a bit speculative, and I would suggest removing it.

6. Line 85: The authors claim they identified bacillamide family with potent cytotoxic activity, however, they show cytotoxic activity for only 2 members of the family out of 4. This should be re-phrased.

7. Lines 120-151: This paragraph contains too much information on methodology that is better suited for Methods section. Although some parts could remain in the main text, I recommend re-writing and moving part of the information to the Methods.

8. Line 164: It is hard to follow the text when the compounds' names are abbreviated to numbers, especially since the names are not too complex. I would suggest coming up with other abbreviations that would remind of the compound the authors are referring to. This would facilitate greatly reading of the article.

9. Line 199: The authors don't mention the antibiotic effect of tryptamine. Was this tested?

10. Line 271: Authors say: "inhibiting growth with IC50-values between 2 and 27 μ M across all cell lines". It should be specified: "inhibiting growth with IC50-values between 2 and 27 μ M across all tested cell lines”.

Reviewer #2 (Remarks to the Author):

In this manuscript, the authors discuss the cytotoxic effects of bacillamide D, a compound produced by *Bacillus cereus* DSM 28590 isolated from mouse intestine.

I believe the manuscript is quite well written and I have not found any elements that would require further clarification to me. Everything is very clear and I consider that the results presented sustain the conclusions derived from it. I have actually enjoyed the reading and found it quite interesting.

I only have a couple of small comments:

- I have seen that *Bacillus cereus* and *B. cereus* are used indistinctively over the whole text. You just have to write the name in full the first time and the rest of times it has to go in abbreviated form. Please amend that.

-The footnotes for Figure 4 seem a bit too brief, if you could add a few more words, not to explain results, but maybe to do it a bit more descriptive, like in figures 1 and 2, that would be good.

To
Communications Biology

Prof. Dr. rer. nat.

Tobias A. M. Gulder

Bearbeiter:

Telefon: 0351 463-34494

Telefax: 0351 463-35506

E-Mail: tobias.gulder@tu-dresden.de

Dresden, February 19th 2024

Dear *Communication Biology* Editorial Team, Dear Dr. Sabine Leanti La Rosa,

We herewith submit our revised article ‘*Bacillamide D produced by Bacillus cereus from the mouse intestinal bacterial collection (miBC) is a potent cytotoxin in vitro*’ authored by Maximilian Hohmann, Valentina Brunner, Widya Johannes, Diminik Schum, Laura M. Carroll, Tianzhe Liu, Daisuke Sasaki, Johanna Bosch, Thomas Clavel, Stephan A. Sieber, Georg Zeller, Markus Tschurtschenthaler, Klaus-Peter Janßen and myself for publication in *Communications Biology*.

We are very thankful for the fast and thorough reviewing process and thank the reviewers and the editorial team for their valuable comments helping us to further improve our manuscript. In response to the issues raised, we have altered the manuscript as listed below:

- **Reviewer 1, major point 1:**

The results on cytotoxic effect of bacillamide D on tumor organoids are promising, however, the study would greatly benefit from in vivo experiments (e.g. administration of bacillamide D to mice with either colorectal cancer or any other tumor model the authors consider appropriate). If the effect persists, it would reinforce the conclusions and increase the relevance of the results. Without these experiments, the study lacks strength.

We agree with the reviewer that having data from mouse experiments to further characterize potential effects of bacillamides *in vivo* is of high interest and would constitute an additional piece of evidence to decipher the role of these compounds in a living organism. However, due to regulatory restrictions and regulations, both EU-wide and national in Germany, this kind of *in vivo* experiment requires prior written approval by the authorities (District Gov. of Upper Bavaria, for TUM), and justification according to the 3R principles for ethical use of animals. In our experience over the last decade, the time to get an approval ranges from 6 to 9 months, sometimes even longer if there are requests from the authorities (followed, of course, by the time required to carry out the actual experiments in mice). Even though we fully agree with the reviewer that it would be highly interesting to have an *in vivo* validation model, we therefore

conclude that it is currently not feasible for us to provide such data within a reasonable timeframe. However, we are planning to include bacillamide profiling, along with a broader selection of other compounds, in a future study in mice *in vivo*. We hope for your understanding.

- **Reviewer 1, major point 2:**

Line 115 and 326: The manuscript is missing information on how the bac was identified and the analysis itself: a) What were the parameters of the analysis? b) What was the output of initial search, or in other words was bac the only BGC that could be identified? It is hard to understand how the authors mined the data and how it led to bac being selected for further analysis. Would this workflow be something that others should explore, or is the identification of bac just one lucky event?

Thank you very much for pointing this out. We agree with the reviewer and have added a short additional statement to the main text of the manuscript (page 5, lines 114-120), provided an additional figure in the ESI (see Figure S9), and included more details on the analysis in the method section (page 14, lines 340-347; please see respective comments in the annotated revised version of the manuscript file).

- **Reviewer 1, major point 3:**

Line 185: It is unclear why the authors focus on the bacillamide C and not A or B. The authors state that they chose bacillamide C because of its altered lipophilicity. However, are there previous studies to support this choice? In fact, authors mention that bacillamide A was previously reported to have algicidal activity, together with the bacillamide D they identified, which would make one think the authors would test the bacillamide A as well, but this is not the case.

Thank you for this comment. The most important compound tested in our study is bacillamide D, as this is the only compound produced from this pathway in our experiments, both, by the natural producer and the heterologous host. To provide some additional insights on one more member of the bacillamide natural product family we decided to also produce bacillamide C by semi-synthesis starting from bacillamide D. We thought this would be of particular interest due to the different lipophilicity of these compounds (free amine in bacillamide D versus amide in bacillamide C). In addition, bacillamide C is the only natural analog that is readily accessible by semi-synthesis from bacillamide D in a single *N*-acetylation reaction. Furthermore, such an *N*-acetylation reaction might also be of relevance *in vivo*, due to non-enzymatic acetylation, e.g., with cellular acetyl-CoA. We have added a corresponding statement to the main manuscript file

(page 7, lines 184-188; please also see respective comments in the annotated revised version of the manuscript file).

- **Reviewer 1, major point 4:**

Could bacillamide D be detected in mouse or human gut and at what concentrations? It would be beneficial for the study and its relevance if the authors provided some information on either bac or bacillamide D presence in the gut microbiota. Could the authors analyze publicly available data (mouse and human) for the presence of bac? How prevalent and abundant it is? Is it possible that the bac product (bacillamide D) is produced in concentrations that would exert effect on the host?

Thank you for your comment. While we have produced bacillamide D by recombinant expression in *E. coli*, we were delighted to also detect the compound (although in minute amounts) in the natural producer. This information was previously hidden in the conclusion section only (page 12, lines 293/294):

HR-MS analysis of lysed cell pellets confirmed the production of bacillamide D by the native producer strain (Figure S8).

We have now added this information to the main text of the manuscript as well (please see page 7, lines 165/166). While we could not detect the presence of bacillamide in mouse or human samples, it is important to note that *B. cereus* is not a major component to the healthy gut microbiome. However, it plays important roles in causing gut-related diseases, such as foodborne emetic intoxication and foodborne diarrheal toxic infection. We have analyzed all publically available genomes for the presence of *bac*, identifying the pathway in 2/3 (!) of all samples. This information can be found in the conclusion section of the main text of the manuscript as follows (page 12, lines 294-299; please also see ESI Figures S7):

It is important to note that the bac BGC occurs frequently within the B. cereus group, being present in about 2/3 of all sequenced B. cereus group genomes (see Figure S7). Members of the B. cereus group are ubiquitously present in natural environments and have been linked to numerous illnesses. As foodborne pathogens, some B. cereus group strains can produce the dodecadepsipeptide cereulide and/or protein-based enterotoxins, allowing them to cause foodborne emetic intoxication and foodborne diarrheal toxic infection, respectively.60–62

The question concerning whether production levels *in vitro* can be sufficiently high to exert effects on the host is very difficult to answer. On the one hand, potential production levels are unknown and will vary significantly depending on the overall gut microbial composition. But titers might actually be high in case of *B. cereus* foodborn infections. On the other hand, concentrations leading to effects on the host are also currently not known and might be impossible to be determined, even with mouse experiments. While compound feeding to mice

can reveal concentrations required for a systemically administered compound, the required concentrations to exert effects in the dynamic situation of the host gut might be significantly lower, e.g., due to local enrichment at a specific infection site or high local concentration during direct microbe/host cell contact. We therefore refrain from commenting on this difficult issue in the manuscript. To make clear that the reported effects were determined in *in vitro* assays, we have amended the title accordingly.

- **Reviewer 1, major point 5:**

In general, the numbers under the Figure 4 and in method section are missing. What is the number of replicates per group for each of the plots? How many times was each experiment performed? What was the statistical test the authors performed and what are the p values? All this information should be included.

We agree with the reviewer and now provide the missing information (page 11, lines 263-273):

Figure 1. Bacillamide D (4) is a potent cytotoxin. **A.** XTT proliferation assays were carried out in technical triplicates. The activity of bacillamides C (4) and D (5) against HEK293 human kidney and HCT116 colon cancer cell lines after 24 h (n=6) and of 4 after 72 h (n=3) is shown with respective IC₅₀-values, derived in GraphPad Prism (8.0.2) after log transformation of raw values, followed by non-linear regression analysis of dose-response inhibition. **B.** HCT116 cells show significantly decreased motility after 12h treatment with Bacillamide C (p=0.0174) or Bacillamide D (p=0.0216) compared to vehicle controls, the effect was even more pronounced after 24h treatment (Bacillamide C, p=0.0035; Bacillamide D, p=0.0034; n=3 assays, mean ± standard deviation is shown, unpaired t-test). **C.** Growth inhibition of Bacillamide D against murine MODE-K cell line (n=4 experiments, normalized against untreated controls). **D, E.** Growth inhibition of Bacillamide D against healthy intestinal organoids (**D**) and intestinal tumor organoids (**E**), each organoid assay was carried out twice with technical triplicates for each assay, normalized against the mean of vehicle treated controls.

- **Reviewer 1, minor point 1:**

The title of the study is somewhat misleading and should be re-phrased: the authors tested only bacillamide D produced by B. cereus from miBC. The other bacillamide (C) was the one identified in marine and not murine strain. This is the reason why the title shouldn't say "Bacillamides produced by Bacillus cereus from the mouse intestinal bacterial collection (miBC) are potent cytotoxins". Alternatively, authors should show that B. cereus from miBC produces bacillamide C as well.

We agree with the reviewer and have amended the manuscript title accordingly.

- **Reviewer 1, minor point 2:**

Unless the study is completed with mouse experiments, the title should reflect that the results are obtained through experiments performed in vitro.

We agree with the reviewer and have amended the manuscript title accordingly.

- **Reviewer 1, minor point 3:**

Line 36. Missing full stop at the end of the sentence.

Thanks! Corrected.

- **Reviewer 1, minor point 4:**

Line 47. This sentence implies that the NPs are somehow produced by interaction of different members of the microbiome. I would suggest saying “However, de-novo-produced NPs from different members of complex microbial ecosystems, such as the human gut microbiome, and their potential effects on host cells have remained poorly understood”.

Thanks! Adjusted accordingly (page 3, lines 58-60).

- **Reviewer 1, minor point 5:**

Lines 59-62. :Missing references. Are there references to support these claims or give an example? If not, it seems a bit speculative, and I would suggest removing it.

Thanks! References added (page 3, lines 62-64).

- **Reviewer 1, minor point 6:**

Line 85. The authors claim they identified bacillamide family with potent cytotoxic activity, however, they show cytotoxic activity for only 2 members of the family out of 4. This should be re-phrased

Thanks! We changed the respective sentence accordingly to (page 3, lines 86-88):

This enabled us to systematically analyze the genomes of isolates for BGCs encoding heterocyclic NPs, which led to the identification of the two members of the bacillamide family as potent cytotoxic NPs.

- **Reviewer 1, minor point 7:**

Lines 120.151. This paragraph contains too much information on methodology that is better suited for Methods section. Although some parts could remain in the main text, I recommend re-writing and moving part of the information to the Methods.

Thank you – we agree! We moved a large part of this section (deleted on page 6, line 141) into the methods part as suggested.

- **Reviewer 1, minor point 8:**

Line 165. It is hard to follow the text when the compounds' names are abbreviated to numbers, especially since the names are not too complex. I would suggest coming up with other abbreviations that would remind of the compound the authors are referring to. This would facilitate greatly reading of the article.

Thanks for this suggestion! In order to make the manuscript easier to follow, we included the compound names in every new paragraph, in addition to their numbers. However, within paragraphs, we would prefer to continue to reference compounds with assigned molecular numbers, as this is commonplace in natural product and chemical literature in general and, in our view, improves readability.

- **Reviewer 1, minor point 9:**

Line 199. The authors don't mention the antibiotic effect of tryptamine. Was this tested?

Thank you. We have added an additional sentence on tryptamine (which we did not further explore) to the manuscript main text (page 9, lines 206-209).

- **Reviewer 1, minor point 10:**

Line 271: Authors say: "inhibiting growth with IC50-values between 2 and 27 μ M across all cell lines". It should be specified: "inhibiting growth with IC50-values between 2 and 27 μ M across all tested cell lines".

Thanks – changed accordingly (page 12, lines 280-281).

- **Reviewer 2, minor point 1:**

I have seen that Bacillus cereus and B. cereus are used indistinctively over the whole text. You just have to write the name in full the first time and the rest of times it has to go in abbreviated form. Please amend that.

Thank you very much for pointing this out – the manuscript was changed accordingly.

- **Reviewer 2, minor point 2:**

The footnotes for Figure 4 seem a bit too brief, if you could add a few more words, not to explain results, but maybe to do it a bit more descriptive, like in figures 1 and 2, that would be good.

We agree with the reviewer and are now providing more detailed information (page 11, lines 263-273; please see above).

We hope we have sufficiently addressed all concerns raised by the reviewers. In case you need any additional information, please let us know. In case you need any additional information, please let us know.

We are looking forward to your final decision on our manuscript.

With best regards,

(Prof. Dr. Tobias A. M. Gulder)

REVIEWERS' COMMENTS:

Reviewer #1 (Remarks to the Author):

I thank the authors for addressing my questions and comments. I greatly appreciate the addition of methodology on how the bac was identified and its prevalence in publicly available genomes. I have no further suggestions.

Reviewer #2 (Remarks to the Author):

I believe that the authors have done a good job replying to the comments, I would recommend acceptance.